# Preparation of CD3 Antibody-Conjugated, Graphene Oxide Coated Iron Nitride Magnetic Beads and Its Preliminary Application in T Cell Separation

**Tianya Liang** [1,†], **Jianxing Li** [1,†], **Xiao Liu** [2], **Zhuang Ma** [2], **Xiaojin Su** [1], **Xiangjiao Meng** [1], **Ziyi Zhanghuang** [1], **Huiqin Wang** [1], **Jintao Li** [1], **Qun Wang** [2,*] and **Minglian Wang** [1,*]

1 Faculty of Environmental and Life Sciences, Beijing University of Technology, Beijing 100124, China; liangtianya@emails.bjut.edu.cn (T.L.); LiJX@emails.bjut.edu.cn (J.L.); SXjin@emails.bjut.edu.cn (X.S.); MengXJ@emails.bjut.edu.cn (X.M.); zhzy@emails.bjut.edu.cn (Z.Z.); wanghuiqin@bjut.edu.cn (H.W.); ljt2008@bjut.edu.cn (J.L.)
2 Faculty of Material and Manufacture Sciences, Beijing University of Technology, Beijing 100124, China; liux@emails.bjut.edu.cn (X.L.); cailiaomazhuang@emails.bjut.edu.cn (Z.M.)
* Correspondence: wangq@bjut.edu.cn (Q.W.); mlw@bjut.edu.cn (M.W.)
† These authors contributed equally to this work.

**Abstract:** Immunomagnetic beads (IMBs) for cell sorting are universally used in medical and biological fields. At present, the IMBs on the market are ferrite coated with a silicon shell. Based on a new type of magnetic material, the graphene coated iron nitride magnetic particle (G@FeN-MP), which we previously reported, we prepared a novel IMB, a graphene oxide coated iron nitride immune magnetic bead (GO@FeN-IMBs), and explored its feasibility for cell sorting. First, the surface of the G@FeN-MP was oxidized to produce oxygen-containing groups as carboxyl, etc. by the optimized Hummers' method, followed by a homogenization procedure to make the particles uniform in size and dispersive. The carboxy groups generated were then condensed and coupled with anti-CD3 antibodies by the carbodiimide method to produce an anti-CD3-GO@FeN-IMB after the coupling efficacy was proved by bovine serum albumin (BSA) and labeled antibodies. Finally, the anti-CD3-GO@FeN-IMBs were incubated with a cell mixture containing human T cells. With the aid of a magnetic stand, the T cells were successfully isolated from the cell mixture. The isolated T cells turned out to be intact and could proliferate with the activation of the IMBs. The results show that the G@FeN-MP can be modified for IMB preparation, and the anti-CD3-GO@FeN-IMBs we prepared can potentially separate T cells.

**Keywords:** graphene oxide; iron nitride magnetic particles; T cell separation; immune magnetic beads



## 1. Introduction

Cell separation is to isolate target cells from multicellular samples, often used in biological research, medical diagnosis, and cell therapy [1,2]. Generally, cell separation methods can be defined as physical and immunomagnetic separation methods. The immune separation method is widely applied for its satisfactory specificity, including immunomagnetic and flow cytometry sorting [3–5]. Immunomagnetic separation is to capture specific cells from mixed cells by immunomagnetic beads (IMBs), essentially magnetic particles coupled with specific antibodies to recognize target cells, aggregating the target cells to the magnetic pole. Compared with flow cytometry separation, immunomagnetic separation is more convenient and efficient, independent of large instruments. After separation, the cells have good biological activity and little impact, which makes it widely used in biological research and clinical application [6–8].

T cells are important human immune cells, accounting for 40–60% of all lymphocytes. T cells function to kill tumor cells, refrain virus replication, and activate macrophages or

neutrophils [9]. T cell separation is very important for cancer treatment and medical research, for instance, the frequently reported chimeric antigen receptor redirected T (CAR-T) therapy or T cell transplantation treatment is based on it [10–12]. Since CD3 is expressed on the membrane surface of mature T cells but not on other lymphocytes, CD3 can be used in IMBs to separate T cells from mixture samples. Furthermore, anti-CD3 antibodies can replace antigen-presenting cells acting as the first signal of T cell activation for proliferating in vitro. The proliferating T cells can be used for immunotherapy [3,13].

Previously, we reported a new type of magnetic material, the graphene-coated iron nitride magnetic particle (G@FeN-MP), which is composed of a multilayer graphene shell bounded around a multiphase iron nitride core [14]. The particles are synthesized using ferrocene by reactive radio-frequency thermal plasma [14,15] and the particle size is about 50 nm.

Iron nitride has a variable valence, including $Fe_4N$, $Fe_3N$, and $Fe_2N$. The magnetic properties and stability of its different valence vary [16]. Compared with ferrite, it has stronger saturated magnetization [17]. Iron nitride is superior in magnetic properties, mechanical strength, corrosion resistance, etc., and used in radiation shields, magnetic recording, medical diagnosis, etc. [18,19].

Graphene is a very light weight material with chemical stability because of its six-membered ring and two-dimensional nanosheet structure. Its large $\pi$ bond also makes it possess excellent conductivity [20]. In recent years, its application in biomedicine has gradually expanded [21]. The stable structural characteristics of graphene include difficulty in interacting with other molecules, an agglomeration tendency, and difficulty in dispersing in various solvents [22], which limits graphene's application. To avoid the disadvantages, much research focuses on the functional modification and derivatives of graphene, such as graphene oxide (GO) [23,24]. The oxidation of graphene destroys part of the conjugate structure and introduces oxygen-containing functional groups [25]. GO retains the excellent properties of graphene and increases its water solubility, biological affinity, and active sites for interaction with other molecules, thus extending its application into the fields of biology and medicine [26]. GO is usually prepared by oxidation of graphite, mostly by Hummers' method [27]. By using an improved Hummers' method, graphene can be prepared more safely and quickly [28], and can even change the proportion of functional groups by controlling the reaction process [29].

So far, most commercially available IMBs are made from ferrites or metal particles such as $Fe_2O_3$ and $Fe_3O_4$ coated with the silica shell. The shell was modified to wear functional groups prior to conjugate antibodies [30,31]. Compared with the silica shelled ferrites, G@FeN-MP we previously reported has much better magnetic responsiveness because of the lower weight of graphene compared to the silica shell. Here, we propose a graphene oxide-coated iron nitride immune magnetic bead (GO@FeN-IMB), which has not been reported previously. GO@FeN-IMB was prepared by graphene oxidation and antibody conjugation (Scheme 1). Our research showed that G@FeN-MP can be applied to prepare IMB, and the anti-CD3-GO@FeN-IMBs can potentially isolate T cells (Scheme 2).

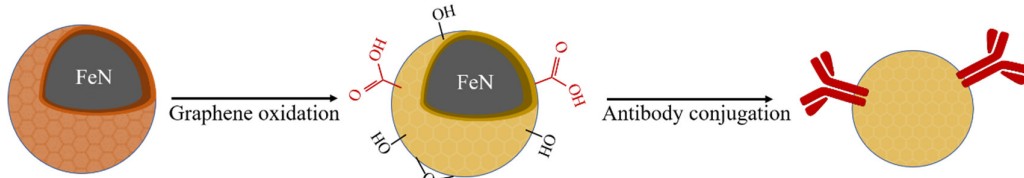

**Scheme 1.** Main workflow of GO@FeN-IMB preparation.

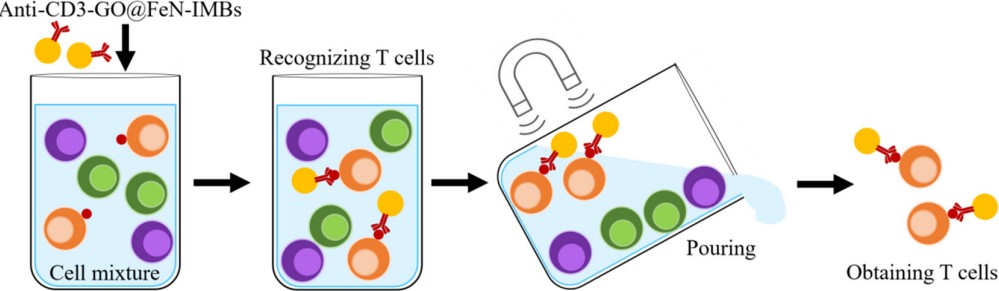

**Scheme 2.** Immunomagnetic separation using anti-CD3-GO@FeN-IMBs.

## 2. Materials and Methods

### 2.1. Preparation of Graphene Oxide-Coated Magnetic Particles (GO@FeN-MP)

The graphene-coated iron nitride magnetic particles (G@FeN-MP) were synthesized by heating ferrocene in an Ar+$N_2$ plasma environment using our RF Thermal Plasma Processing System [15]. Then, the improved Hummer's method was used to oxidize the graphene on the surface of G@FeN-MP [32]. One quarter gram of G@FeN-MP and 0.5 g sodium nitrate ($NaNO_3$) were put into a beaker placed in ice; then, slowly and evenly, 23.0 mL concentrated sulfuric acid ($H_2SO_4$) and 1.0 g potassium permanganate ($KMnO_4$) were added. The mixture was then shaken and dispersed at 35 °C for 30 min in a water bath ultrasonic apparatus, then mechanically mixed for another 30 min to ensure particles dispersed sufficiently. Forty-six milliliters of deionized water was added slowly to initiate the exothermic reaction, followed by heating at 80 °C for 15 min. After the mixture was cooled to room temperature, an appropriate amount of 30% $H_2O_2$ was added to consume the excessive oxidant until air bubbles vanished. Then, GO@FeN-MP was obtained with the help of a magnet after washing with deionized water several times. Finally, it was dried at 55 °C in vacuum drying oven (DRY-BIG, J.P. SELECTA, S.A.) and tested by Fourier transform infrared spectroscopy (FTIR), X-ray photoelectron spectroscopy (XPS), and transmission electron microscopy (TEM).

NaCl, $NaNO_3$, $KMnO_4$, 95% $H_2SO_4$, 30% $H_2O_2$ were purchased from Beijing Chemical Works (Beijing, China).

### 2.2. Homogenization of GO@FeN-MP

One tenth gram of GO@FeN-MP was dissolved in 500 mL deionized water, with about 1/6 total volume of ethanol added to the suspension. The mixture was poured into the feed inlet port of an ultra-high-pressure homogenizer (FB-110Q, Shanghai Litu Mechanical Equipment Engineering Co., Ltd., Shanghai, China). The valve of the machine was regulated to increase the total pressure up to 400 bar. After circulating two or three times in the homogenizer, the mixture was homogenized. Then, the GO@FeN-MP solution was collected. The uniform effect and particles size were determined by Zetasizer Nano (S90, Malvern Panalytical, Malvern, UK), TEM, and an optical microscope.

### 2.3. Conjugating GO@FeN-MP with Proteins (BSA or Antibodies)

The diimide-activated amidation reaction [33] was used to conjugate GO@FeN-MP with proteins by condensation reaction of the generated carboxyl groups on the particles with amino groups of the proteins. The condensation agents, 1-(3-Dimethylaminopropyl)-3-ethylcarbodiimide hydro (EDC·HCl) and N-hydroxysuccinimide (NHS) were obtained from ShangHai Medpep Co., Shanghai, China, Ltd. and phosphate buffer saline (PBS) was from GIBCO (Thermo Fisher Scientific Inc., Waltham, MA, USA). The proteins here were perhaps BSA or antibodies. Specifically, 16 mg EDC·HCl and 24 mg NHS were added to a 1 mL magnetic particles solution (10 mg/mL) to activate carboxyl groups in a horizontal shaker at 100 rpm for 15 min at room temperature. The magnetic particles were drawn out of the liquid with a magnet. The particles were washed with PBS buffer two or three times, 20 μL proteins (2 mg/mL) added, and blended in a thermostatic mixer at 180 rpm for at

least two hours at room temperature. The conjugated magnetic particles were sucked out with a magnet and washed with PBS to remove the free proteins. Then, the particles were resolved in PBS buffer at a concentration of about 10–20 mg/mL.

### 2.4. Detection of Coupled Protein Content

BSA (BD Biosciences, Japan) was used to couple GO@FeN-MP for determining the content of coupled protein since the BCA assay method is commonly used for protein concentration detection by detecting the absorbance of the purple-colored complex at 562 nm. After the conjugation of BSA to magnetic particles at different quality proportions following steps in Section 2.4, the amount of proteins coupled with GO@FeN-MP were detected using a multifunctional microplate reader (EnSpire, PerkinElmer, Waltham, MA, USA), and the absorbance value was substituted into the standard curve for calculation as the introduction of BCA protein quantification kit (SoLarbio Life Sciences, Beijing, China). Due to the different quality of GO@FeN-MP, the amount of coupling proteins per milligram GO@FeN-MP was calculated.

### 2.5. Detection of the Bioactivity of the Conjugated Antibodies

HRP-labeled secondary antibodies (BIOSYNTHESIS) were used to couple GO@FeN-MP following the steps in Section 2.3, and the HRP-labeled antibodies conjugated magnetic particles were dispensed in wells of a 96-well plate. A chromogenic solution of 3,3′,5,5′-tetramethylbenzidine (TMB) (DINGGUO CHANGSHENG Biotechnology Co., Ltd. Beijing, China) was used to detect HRP activity by measuring the absorbance at 450 nm. One-percent TMB and 0.5% $H_2O_2$ were added into the substrate buffer to prepare a fresh TMB chromogenic solution. One hundred microliters of the fresh coloring solution per well was added to the coupling group and blank control group (uncoupled GO@FeN-MP), and incubated at 37 °C for 10 min and the absorbance value was read at 450 nm.

In addition, the FITC-labeled secondary antibodies (Jackson ImmunoResearch, West Grove, PA, USA) were coupled with GO@FeN-MP following the steps in Section 2.3, and the fluorescence of the conjugated particles was observed under the fluorescence microscope at blue light band (450–490 nm).

### 2.6. Detection of Bounding Stability of Antibodies with GO@FeN-MP

Here, we used SDS PAGE and Coomassie brilliant blue staining to detect proteins assumed to fall off the magnetic particles under a denatured condition. Three milligrams of antibody-coupled GO@FeN-MP, i.e., IMB, were placed on the magnetic stand to remove the supernatant. A 5× loading buffer was added with the final volume of 100 μL. The mixture was heated at 100 °C for 10 min to denature conjugated proteins, and then the heated supernatant was collected after magnetic separation. Ten percent polyacrylamide gel was prepared with a PAGE gel fast preparation kit (EpiZyme, Shanghai, China), and 20 μL of the sample was added to each well while denatured homologous free protein was added as a positive control. After gel electrophoresis, 50 mL deionized water was added to the submerged gel and boiled for 30 s. After removing the water, the gel was immersed with 50 mL Coomassie brilliant blue working solution (Solarbio Life Sciences, China) and boiled for 30 s and shaken on the horizontal shaker for 5 min. The staining solution was replaced with deionized water and boiled another 30 s, and shaken for 5 min again until the gel decolorized, followed by observation to determine if any band showed.

### 2.7. Isolation of CD3+ Cells from Cell Mixture

Anti-CD3-GO@FeN-IMBs were prepared by coupling GO@FeN-MP with antihuman CD3 antibody (BD Biosciences, San Jose, CA, USA) following the steps in Section 2.3, and diluted to a final concentration at about 10 mg/mL. The human T cell line MT-4 cells and two other types of cells (EC109 and NIH/3T3) were mixed at 1:1:1 to a total of $10^7$ cells. The mixed cells were dispersed in 1 mL PBS buffer containing 1% FBS. Two hundred microliters of anti-CD3-GO@FeN-IMBs was added to the cell mixture in a tube and gently shaken for

10 min at room temperature. The mixture was then placed on a magnetic stand for 10 min and the supernatant removed. Cells trapped by the magnet were washed with PBS buffer three times and suspended with Roswell Park Memorial Institute (RPMI) 1640 culture medium (GIBCO). Cells were cultured at 37 °C and observed daily thereafter. When cells proliferated, the IMBs were separated from the cells using a magnetic stand.

### 2.8. Statistical Analysis

Mean and standard error (SE) were calculated for each parameter. Results were expressed as mean ± SE of multiple determinations. Comparisons between levels were evaluated by one-way ANOVA. A statistically significant difference was assumed for $p < 0.05$.

### 3. Results and Discussion

#### 3.1. Characterization of the Graphene Oxide-Coated Magnetic Particles (GO@FeN-MP)

The functional groups on the surface of the particles were analyzed by FTIR. Graphene had a strong and wide spectrum of light absorption; the particles before oxidation had no peak when detected by infrared spectrum. After oxidation, the functional group peaks appeared, which was caused by the strong oxidation regents destroying the conjugate structure.

Figure 1a shows that the main characteristic peaks with different intensity appear at around 3000, 1735, 1620, 1380, 1218, and 1050 cm$^{-1}$. The groove around 3000 cm$^{-1}$ and the peak at 1735 cm$^{-1}$, respectively, are stretching vibration peaks of the O-H bond and C=O bond of carboxyl, and the 1620 and 1380 cm$^{-1}$ peaks are of the O=C-O bond of carboxylate, showing our GO@FeN-MP contained carboxyl groups. Peaks at 1218 and 1050 cm$^{-1}$ are the C-O-C bond.

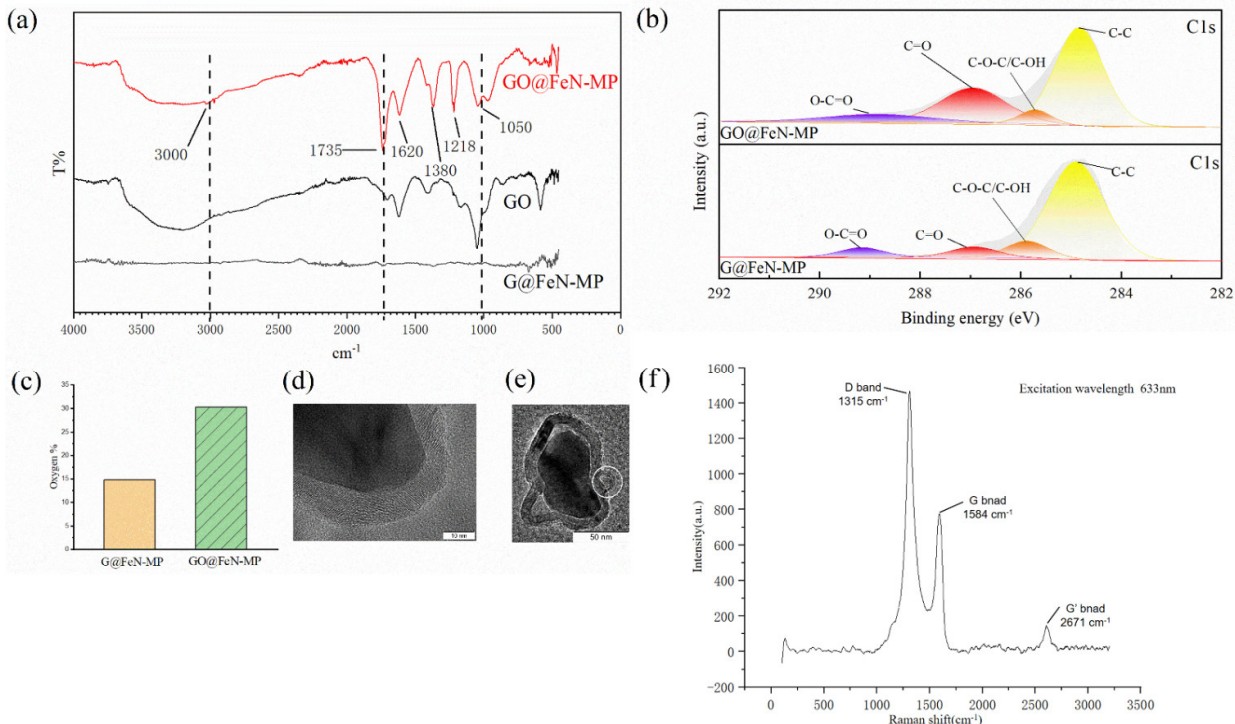

**Figure 1.** The oxidation effect of G@FeN-MP: (**a**) infrared spectrum of GO@FeN-MP, graphene oxide, and G@FeN-MP; (**b**) XPS spectra of GO@FeN-MP and G@FeN-MP; (**c**) differences in oxygen% of GO@FeN-MP and G@FeN-MP; (**d**) TEM image of GO@FeN-MP; (**e**) TEM image of GO@FeN-MP; (**f**) surface-enhanced Raman spectroscopy of GO@FeN-MP.

In FTIR, there are various peaks shifted from those of GO with large intensity changes. The red shift is due to formation of many polar groups such as hydroxyl, carbonyl, and carboxyl groups in the oxidation process of the GO@FeN-MP sample. The strong elec-

tronegativity of these groups reduces the bond energy between C-C bonds of graphene oxide on the surface of GO@FeN-MP, resulting in the red shift of the spectrum.

Purity GO (Chengdu Organic Chemicals Co. Ltd., Chengdu, China) was used as positive control. The characteristic peaks of GO@FeN-MP were basically consistent with control, and the functional groups mentioned above shown in Figure 1a also conformed to the characteristics of GO [34], which proved that our preparation method was successful.

XPS was performed to overview the extensive characterization of surface functional groups. Figure 1b showed that GO@FeN-MP had a higher intensity of oxygenic groups, especially carbonyl groups (C=O) at 286.91 eV. The percentage of oxygen in GO@FeN-MP was greater than that in G@FeN-MP (Figure 1c), which also supported the formation of oxygen functional groups.

TEM and enhanced Raman spectra revealed that the GO@FeN-MP structure was intact. The surface-enhanced Raman spectra of GO@FeN-MP (Figure 1f) show that there are D and G peaks at 1315 and 1584 cm$^{-1}$, which are the characteristic peaks of graphene oxide, and there is no characteristic peak of FeN$_4$ in the spectra, so the GO@FeN-MP structure was intact. TEM images also support this conclusion (Figure 1d). The GO@FeN-MP's cores were surrounded by a 10 nm thin layer of graphene oxide shell, but the drying step made the GO@FeN-MP gained difficult to disperse. Some particles aggregated, forming large particles that required homogenization.

In addition, it was found that the GO shell of some G@FeN-MPs did not wrap the core tightly. As shown in the Figure 1e, there was a diabiosis on the right side of the particles, where the graphene shell was damaged, and the strong oxidant solution might enter the iron nitride core causing corrosion. A similar image can be seen in the red circle shown in Figure 2b. It reminded us that the reaction temperature and time must be controlled to prevent the oxidation from going too far.

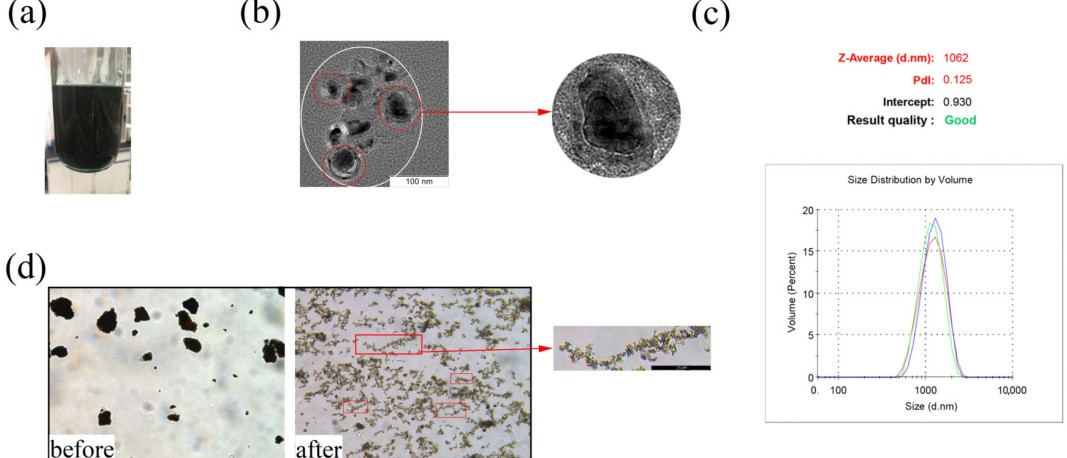

**Figure 2.** Homogeneous effect of GO@FeN-MP: (**a**) visual observation of homogenized GO@FeN-MP; (**b**) TEM image of homogenized GO@FeN-MP; (**c**) average particle size of homogenized GO@FeN-MP; (**d**) GO@FeN-MP under microscope before and after homogenization.

### 3.2. Homogeneous Effect of GO@FeN-MP

The GO@FeN-MP tended to aggregate, impeding further application, so it is necessary to be homogeneously dispersed. The homogenized GO@FeN-MP suspension was stable by visual observation, without particle sensation (Figure 2a), and the homogeneity kept on as long as a month. After standing a long time, the homogenized GO@FeN-MP gradually settled due to the gravity, but could be restored to a uniform solution by mechanical mixing. The particle size of G@FeN-MPs was about 50 nm. After homogenization, the GO@FeN-MP did not completely disperse into independently free small particles, but formed into coacervates. As shown under TEM in Figure 2b, small particles of complete core–shell

structure aggregated into larger particles, and the particle size analyzer recognized the agglomerated small particles shown in the white circle as one large particle, averaging 1062 nm (Figure 2c). Compared with the sample before homogenization, the homogenized GO@FeN-MP had better dispersibility, and only showed slight and loose aggregation under the microscope (Figure 2d). Unexpectedly, we found that small GO@FeN-MPs linked into chains of magnetic beads, perhaps due to magnetization by the geomagnetic field. Homogenized GO@FeN-MP could be used for subsequent experiments only by blowing and mixing without ultrasonic treatment.

### 3.3. Coupling Effect of GO@FeN-MP with Proteins

We used BSA as a common protein to conjugate with GO@FeN-MP to determine the coupling efficiency. As shown in Figure 3a, the amount of bound proteins increased with the increasing protein to GO@FeN-MP ratio. The maximum BSA protein coupled was $21.21 \pm 0.42$ µg per milligram of GO@FeN-MP, which was significantly higher than other ratios of BSA to GO@FeN-MP ($p < 0.05$). It showed that the best coupling efficiency was achieved at the mass ratio of BSA to GO@FeN-MP of 1:50. It was roughly estimated that the binding ratio of antibodies with high concentration of BSA to GO@FeN-MP was between 1:50 and 3:50, on account of the molecular weights of BSA and IgG antibodies, which are 67 and 150 kDa, respectively.

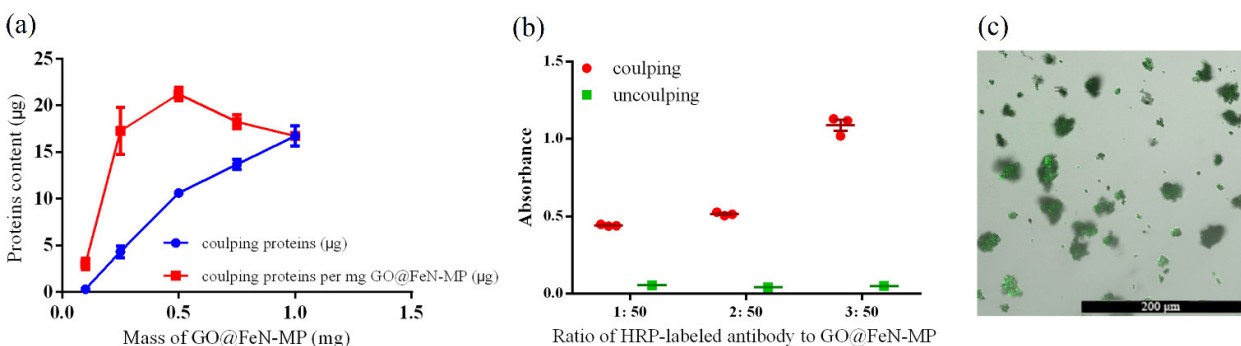

**Figure 3.** The coupling efficiency of GO@FeN-MP: (**a**) binding protein content of GO@FeN-MP; (**b**) detection of HRP-labeled antibody-coupling GO@FeN-MP; (**c**) FITC-labeled antibody-coupling GO@FeN-MP observed under a fluorescence confocal microscope.

In this research, we used the second antibody to further verify the coupling effect. First, an HRP-labeled antibody was coupled with GO@FeN-MP and the substrate TMB was added to the solution following Section 2.3. The measured absorbance value of coupling groups was significantly higher than the uncoupling groups, which was proved by repeated experiments (Figure 3b), indicating that the HRP-labeled antibody coupled still remained active.

In addition, another secondary antibody, i.e., the FITC-labeled antibody was coupled to the GO@FeN-MP, and green fluorescence could be observed on the surface of GO@FeN-MP under a fluorescence microscope (Figure 3c). All the results indicated that the coupling method we used did not significantly impair the bioactivity of the conjugated antibodies.

To examine the stability of the conjugation, the Ab-GO@FeN-MP was denatured at boiling temperature. No protein was detected in the solution when removing the GO@FeN-MP, showing that the denaturing conditions did not make the proteins fall off the GO@FeN-MP, which means the protein binding was stable.

### 3.4. Separation of MT-4 Cell with Anti-CD3-GO@FeN-IMBs

The anti-CD3-GO@FeN-IMBs we prepared were mixed and incubated with a cell mixture including MT-4 cells, a kind of T cell line that tends to agglomerate during proliferation.

The target cells captured by anti-CD3-GO@FeN-IMBs were collected into cell dish and cultured. Its proliferation was observed for a week (Figure 4a). The isolated cells were

observed to be linked with IBMs (day 0). After 24 h of culture, cells and IMBs got together and turned to be much larger agglomerates.

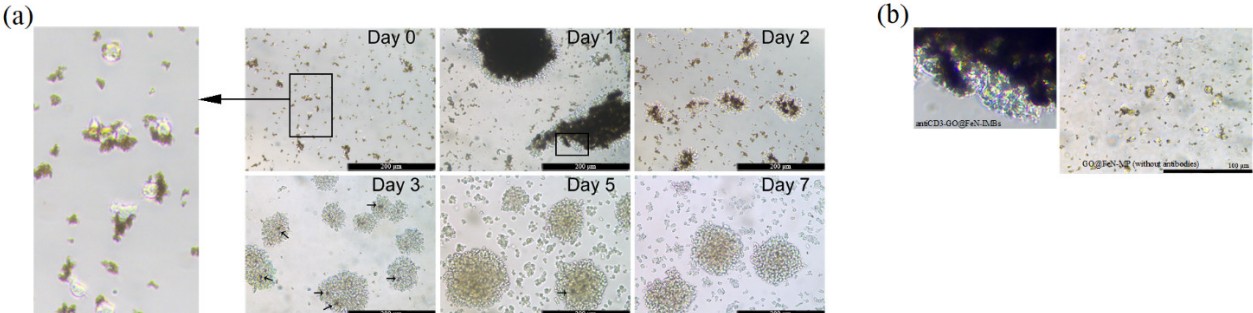

**Figure 4.** Proliferation of MT-4 cells with anti-CD3-GO@FeN-IMBs: (**a**) MT-4 cells observed under a microscope on day 0, days 1, 2, 3, 5, and 7; (**b**) coculture of MT-4 cells with anti-CD3-GO@FeN-IMBs or GO@FeN-MP (without antibodies).

To investigate whether it was the magnetism of IMBs or CD3 antibody bridges that aggregated the IMBs and cells into a huge mass, we cocultured the GO@FeN-MP (without CD3 antibody) with MT-4 cells. Figure 4b shows that the aggregation of cells and IMBs did not occur, which excluded the spontaneous aggregation due to magnetization, indicating the agglomeration (Figure 4a, day 1) might be attributed to the CD3 antibody bridges.

After massive proliferation, most IMBs were removed by magnetic separation. Sporadic dark dots can be seen in the cell cluster on day 3 and 5, which were residual IMBs. Although it was reported that GO has a dose-dependent damage effect on eukaryotic cells [35], the dosage of IMBs we used did not cause significant cell damage. The morphological characteristics of the isolated cells were intact and the cells continued to grow normally, suggesting that anti-CD3-GO@FeN-IMBs we prepared are appropriate for separating T cells.

Although anti-CD3-GO@FeN-IMBs we prepared successfully isolated T cells from the cell mixture, the IMBs would perform better if customized antibodies were used, since the commercialized antibody we used contained a high content of BSA, which competitively inhibits antibody conjugation.

## 4. Conclusions

GO@FeN-MPs derived from G@FeN-MPs carry carboxyl groups, which can be used for protein coupling. After homogenization, the particle size become uniform, and the GO shell still keeps effective carboxyl groups. Antibodies are successfully conjugated with GO@FeN-MPs, thus IMBs can be prepared.

The anti-CD3-GO@FeN-IMBs we prepared can be used for T cell separation and the captured T cells can normally proliferate, which is convenient for downstream experiments, indicating that the GO@FeN-IMBs are applicable in the field of cell separation. Based on the superiority of the new type magnetic material, we believe that GO@FeN-IMB products formed in the future will be more efficient on the condition that customized antibodies with less BSA-concentration be used for coupling reaction.

Promisingly, we think the GO@FeN-IMBs may not only be used for cell separation, but also for hooking specific microorganisms or biomacromolecules dependent on the specific antibodies conjugated.

**Author Contributions:** Formal analysis, X.S.; data curation, X.L., Z.M., X.M., Z.Z., H.W., J.L. (Jintao Li); writing—original draft preparation, T.L.; writing—review and editing, J.L. (Jianxing Li); project administration, Q.W., M.W. All authors have read and agreed to the published version of the manuscript.

**Funding:** This research was funded by the Natural Science Foundation of Beijing Municipality (grant number 7182011) and National Key Research and Development Project (grant number 2016YFB1200602—37).

**Institutional Review Board Statement:** Not applicable.

**Informed Consent Statement:** Not applicable.

**Conflicts of Interest:** The authors declare no conflict of interest.

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
