# Peer review of "Preparation of CD3 Antibody-Conjugated, Graphene Oxide Coated Iron Nitride Magnetic Beads and Its Preliminary Application in T Cell Separation"

_magnetochemistry, doi:10.3390/magnetochemistry7050058_

Round 1
Reviewer 1 Report
Journal: Magnetochemistry (ISSN 2312-7481)
Manuscript ID: magnetochemistry-1158572
Type: Article
Title: Preparation of CD3 antibody-conjugated, graphene oxide coated iron nitride magnetic beads and its preliminary application in T cell separation
Authors: Tianya Liang , Jianxing Li , Xiao Liu , Zhuang Ma , Xiaojin Su , Xiangjiao Meng , Ziyi Zhanghuang , Huiqin Wang , Jintao Li , Minglian Wang * , Qun Wang *
The manuscript deals with both preparation and characterisation of antibody-conjugated inorganic (magnetic) beads for T cell separation.
The authors report interesting results and propose quite clear interpretations as well, with good referencing.
In my opinion the manuscript can be accepted for publication as is in Magnetochemistry.
Author Response
Thank you very much for your acceptance of the research of our team, and thank you for all your comments, which is very helpful to improve the quality of our papers.

Reviewer 2 Report
Liang et al. report synthesis of GO-coated FeN nanobeads toward T cell sorting applications. While experimental data are generally supporting the authors’ main claim, the manuscript should be improved.
In FTIR, there are various peaks shifted from those of GO with the large intensity changes. But, there are no explanations on the shifts and intensity changes. Need detailed discussion on this part.
Authors mentioned that the 1218 cm-1 and 1050 cm-1 peaks are for the C-O-C bond of by-products. What are the by-product materials? Does it mean that the samples have impurity, but authors did not conduct purification and do not recognize the source of impurity?
“TEM revealed that the GO@FeN-MP structure was intact (Figure 1d)”. Authors did not show any X-ray or Raman experiments to prove that the structure is not damaged. One TEM micrograph is not sufficient to support this claim.
Provide VSM data of FeN and GO@FeN for basic magnetic properties (Ms and magnetic hysteresis).
- There are many grammatical errors. I strongly suggest authors to have professional English proofread service.
- Minutes and min are mixed. Keep consistency.
- Fig 1a caption only mentions graphenes. Revision is necessary.
- Caption for Fig 1e is missing.
- Quality of some Figures are not sufficient for publication. Authors should redraw Fig 2(c) using the raw data.
- Fig 2b shows about 250-300 nm size for the agglomerated particles, but Fig 2d says over 1000 nm. Authors should clarify this mismatch.
- Captions for scale bar in Fig 2d and Fig 4 are missing.
- Scale bar is missing in Fig 3c.
- In Fig 4, “p” of proliferation should be capitalized.
Author Response
Thank you very much for your acceptance of our research, and thank you for all your comments, which is very helpful to improve the quality of our papers.Our reply to your question is attached

Round 2
Reviewer 2 Report
Authors addressed previously raised concerns and I recommend to accept this manuscript for publication in Magnetochemistry.